# *Shigella flexneri* Outbreak at a Rehabilitation Center: First Report from Saudi Arabia

**DOI:** 10.3390/healthcare13090971

**Published:** 2025-04-23

**Authors:** Khalifa Binkhamis, Sarah Alangari, Fatema Juma, Sahar Althawadi, Ahmed A. Al-Qahtani, Marie Fe F. Bohol, Fatimah S. Alshahrani, Fawzia Alotaibi

**Affiliations:** 1Department of Pathology, College of Medicine, King Saud University, Riyadh 11461, Saudi Arabia; sarah.alangari21@gmail.com (S.A.); fatema_ajj@hotmail.com (F.J.); ofawzia@ksu.edu.sa (F.A.); 2Microbiology Unit, Medical Laboratories and Blood Bank, King Saud University Medical City, King Saud University, Riyadh 11461, Saudi Arabia; 3Department of Pathology, Bahrain Defence Force Hospital-Royal Medical Services, Riffa 928, Bahrain; 4Section of Microbiology, Department of Pathology and Laboratory Medicine, King Faisal Specialist Hospital and Research Center, Riyadh 11211, Saudi Arabia; sthawadi@kfshrc.edu.sa; 5Department of Infection and Immunity, King Faisal Specialist Hospital and Research Center, Riyadh 11211, Saudi Arabia; aqahtani@kfshrc.edu.sa (A.A.A.-Q.); mbohol@kfshrc.edu.sa (M.F.F.B.); 6College of Medicine, King Saud University, Riyadh 11362, Saudi Arabia; falshahrani1@ksu.edu.sa; 7Division of Infectious Diseases, Department of Internal Medicine, King Saud University Medical City, King Saud University, Riyadh 11362, Saudi Arabia; 8Infection Prevention and Control Department, King Saud University Medical City, King Saud University, Riyadh 11362, Saudi Arabia

**Keywords:** outbreak, *Shigella* spp., rehabilitation center, Saudi Arabia

## Abstract

**Background: ***Shigella flexneri* is a major cause of shigellosis in developing regions and is known to cause outbreaks in institutional settings. Transmission occurs via the fecal–oral route. It invades intestinal epithelial cells, causing diarrhea, systemic symptoms, and complications such as hemolytic uremic syndrome. This study aimed to characterize the clinical presentation, administered treatment, infection outcomes, and infection control measures during a local *S. flexneri* outbreak at a rehabilitation center. **Methods**: This case series at King Saud University Medical City (Oct–Dec 2024) investigated *S. flexneri* infections from a rehabilitation center. Stool and blood samples were cultured and analyzed using microbiological methods. Molecular studies were used to verify the genetic linkage between the isolates and to study their virulence genes. **Results**: Four cases of *S. flexneri* were included, involving patients with various comorbidities, residing in a rehabilitation center, and presenting with symptoms like fever and diarrhea. Laboratory investigations revealed leukocytosis, electrolyte imbalances, and elevated inflammatory markers. Imaging studies showed findings consistent with colitis in two cases. Patients were managed with IV fluids and targeted antibiotics, leading to symptom resolution. Molecular studies confirmed the genetic relatedness between the *S. flexneri* isolates, with virulence genes indicating cellular invasion and inflammation as primary drivers of disease severity. Outbreak management comprised contact isolation, environmental disinfection, and education. **Conclusions**: *S. flexneri* outbreaks in long-term care facilities pose challenges among bedbound patients. Diapers may facilitate transmission, and infections may cause severe complications. Robust infection control, identifying outbreak sources, and strengthening prevention strategies are essential to protect vulnerable populations.

## 1. Introduction

*Shigella* species (spp.) are Gram-negative facultative anaerobic bacteria that cause shigellosis, with *Shigella flexneri* being a major pathogen in developing regions. It contributes to an estimated 165 million infections and 1.1 million deaths annually, primarily among children [1,2]. Transmission occurs via fecal–oral routes through contaminated food and water, exacerbated by poor sanitation in low-income settings [3,4]. *S. flexneri* invades intestinal epithelial cells using Ipa proteins [1,5], causing gastrointestinal symptoms and complications such as hemolytic uremic syndrome (HUS) [6,7]. *S. flexneri* is classified into 15 serotypes with geographic variation [8]. Epidemiological trends show *S. sonnei* surpassing *S. flexneri* in some regions [9], influenced by factors like climate change, urbanization [10,11], and sexual practices [12].

*S. flexneri* infection can manifest with a spectrum of clinical features. The primary manifestations are diarrhea, often severe and bloody (dysentery), accompanied by abdominal cramps and tenesmus, typically occurring 1–3 days post-exposure [6,13]. Systemic symptoms include fever, malaise, chills, and fatigue [6,14]. Multiple complications may ensue with *S. flexneri* infection. These include dehydration in children/elderly, reactive arthritis, and septicemia in immunocompromised patients [15,16,17]. Chronic outcomes include prolonged diarrhea, post-infectious irritable bowel syndrome, and growth retardation in children due to persistent malnutrition [6,18].

Stool culture remains the gold-standard method for detecting *Shigella flexneri*, using selective media such as Hektoen (HEK) agar or Xylose Lysine Deoxycholate (XLD) agar to isolate the pathogen [19]. Suspected colonies undergo biochemical testing for confirmation [20]. However, culture methods can be slow. Polymerase Chain Reaction (PCR) provides rapid and specific detection by targeting virulence genes like *ipaH* [21]. Multiplex PCR can identify *Shigella* spp. within an hour, making it useful for clinical diagnostics [22]. Moreover, immunological assays such as enzyme-linked immunosorbent assay and lateral flow tests use antibodies to detect *Shigella* spp. antigens in stool samples [23,24].

Supportive care is vital in shigellosis treatment, mainly to prevent dehydration. Oral rehydration solution is recommended for mild to moderate cases, while severe dehydration may require intravenous fluids [3]. Antibiotic therapy is necessary for moderate to severe cases. Fluoroquinolones, like ciprofloxacin, have been used as first-line treatments, but increasing resistance demands careful antibiotic selection based on local patterns [3,25]. In resistant cases, azithromycin or ceftriaxone may be alternatives, with azithromycin preferred for children [25,26]. Multidrug-resistant *Shigella flexneri* requires susceptibility testing to guide therapy [27].

*S. flexneri* outbreaks have been reported in various institutional settings, often facilitated by close contact and foodborne transmission. In 2019, an outbreak in a Chinese mental healthcare center was linked to person-to-person spread or contaminated food and was controlled through quarantine and improved hygiene [28]. Another outbreak of a multidrug-resistant strain in a children’s welfare institute was traced to a newly admitted child, with transmission exacerbated by crowded living conditions [29]. In a hospital in South East England, transmission among patients and staff required strict infection control measures [30]. Similarly, a long-stay psychiatric center experienced a sustained outbreak of *Shigella flexneri* 4a, during which antibiotic prophylaxis was used. This resulted in significant strain variation and resistance, emphasizing the importance of infection control measures over the use of antibiotic prophylaxis [31].

Strict hygiene practices are essential for preventing *Shigella* spp. transmission. Regular handwashing with soap and water is crucial, especially after restroom use and before handling food [32]. Alcohol-based hand sanitizers should be encouraged in healthcare settings where soap and water are unavailable. Proper food safety measures, such as hygiene training for food handlers, cooking at safe temperatures, and preventing cross-contamination, can help reduce outbreaks [33]. Surveillance systems are necessary for early detection, rapid isolation, and outbreak response [33]. Education and training programs for healthcare workers and the public can improve awareness of hygiene and transmission risks [32,33]. Environmental cleaning in healthcare facilities where patients with diarrhea are treated further minimizes contamination risks [34]. This study aimed to characterize the clinical manifestations, administered treatment, infection outcomes, and infection control measures during a *Shigella flexneri* outbreak at a rehabilitation center. In addition, we sought to investigate the genetic relatedness of the isolates through molecular epidemiological methods and to identify the isolates’ virulence genes.

## 2. Materials and Methods

### 2.1. Study Design

We conducted a case series on *S. flexneri* from 1 October 2024 to 31 December 2024 at King Saud University Medical City (KSUMC), Riyadh, Saudi Arabia. We included all patients with suspected *S. flexneri* infection who presented with symptoms of fever and/or diarrhea to the KSUMC emergency department (ED) from the rehabilitation center during the outbreak period and excluded those with negative culture results for *Shigella* spp. The laboratory information system and electronic medical records were retrospectively and prospectively reviewed for each patient to collect the data, which included demographic details, clinical manifestations, laboratory results, management, and outcomes.

### 2.2. Microbiological Diagnosis

#### 2.2.1. Stool Culture

Stool samples were collected in a sterile, wide-mouth container and immediately sent to the bacteriology laboratory. The samples were inoculated on XLD agar, Campylobacter (Campy) agar, sorbitol MacConkey agar (prepared by Saudi Prepared Media Company, Riyadh, Saudi Arabia), Cefsulodin–Irgasan–Novobiocin (CIN) agar, and selenite F broth (prepared by Second Advanced Medical Company, Riyadh, KSA). The XLD agar, CIN agar, sorbitol MacConkey agar plates, and selenite F broth were incubated at 35 ± 1 °C in ambient air for 24 h, while the Campy agar plate was incubated at 42 ± 1 °C in a microaerophilic environment for 48 h. After incubation for 18–24 h, a subculture from selenite F broth was inoculated on an HEK agar plate and incubated for another 24 h at 35 ± 1 °C in ambient air. Growth on plates was examined after 24 h and 48 h incubations. Identification and susceptibility testing were achieved via the DxM MicroScan WalkAway ID/AST System using the NBC50 panel (Beckman Coulter, Brea, CA, USA), and speciation as *Shigella flexneri* was achieved using the Wellcolex™ Color *Shigella* Kit (Thermo Scientific™ Remel™, Waltham, MA, USA).

#### 2.2.2. Blood Culture

Two sets of blood cultures were collected using an aseptic technique from both arms for all included patients. The bottles were incubated using the blood culture continuous monitoring system (BACT/ALERT^®^ VIRTUO^®^ | bioMérieux, Marcy-l’Étoile, France) for a maximum duration of 5 days. At any time during that time, if a bottle was flagged as positive by the instrument, a Gram stain was performed, and the nurse in charge was immediately informed of the result. The bottles were subcultured on sheep blood agar, chocolate agar, and MacConkey agar plates (prepared by Saudi Prepared Media Company, Riyadh, Saudi Arabia). The sheep blood agar and chocolate agar plates were incubated at 35 ± 1 °C with 5% CO_2_, while the MacConkey plates were incubated at 35 ± 1 °C in ambient air. The plates were examined after overnight incubation. Subsequently, identification and susceptibility testing were performed using the DxM MicroScan WalkAway ID/AST System via the NBC50 panel (Beckman Coulter, CA, USA). Further speciation of *S. flexneri* was performed using the Wellcolex™ Color *Shigella* Kit (Thermo Scientific™ Remel™, MA, USA).

### 2.3. Molecular Diagnosis

Isolates were referred out for molecular epidemiological studies and virulence gene detection.

#### 2.3.1. Pulsed-Field Gel Electrophoresis (PFGE)

A single bacterial colony was cultured overnight in 3 mL of tryptic soy broth (Sigma, St. Louis, MO, USA, Catalog No. 22091) at 37 °C in a shaker incubator. The cells were then harvested via centrifugation at 3000 rpm for 10 min, and the supernatant was discarded. The pellet was resuspended in SE buffer [25 mM EDTA (pH 8.0), 75 mM NaCl (pH 8.0)], with the final volume adjusted based on an OD of 1.4 at 610 nm.

For plug preparation, equal volumes of the bacterial suspension and 2% low-melting-point agarose (Invitrogen, Carlsbad, CA, USA, Catalog No. 15517-022) were mixed and pipetted into a reusable plug mold. The mixture was allowed to solidify for 30 min at 4 °C. The bacterial plugs were then subjected to lysis through overnight incubation at 55 °C in a water bath with 1 mL of lysis buffer [50 mM Tris-HCl (pH 8.0), 50 mM EDTA (pH 8.0), 1% sodium lauroyl sarcosine] supplemented with 1 mg/mL Proteinase K (Thermo Fisher, San Francisco, CA, USA, Catalog No. 25-530-015).

The plugs were washed once in 5 mL of sterile distilled water for 5 min at room temperature, followed by four washes in 3 mL of TE buffer [10 mM Tris-HCl (pH 8.0), 1 mM EDTA (pH 8.0)] at room temperature, each lasting 30 min. A 3 × 5 mm plug was equilibrated with 1X CutSmart buffer for 30 min at 4 °C before being digested overnight at 37 °C with 50 units of NotI (New England Biolabs, Ipswich, MA, USA, Catalog No. R0189L) and/or SpeI (New England Biolabs, MA, USA, Catalog No. R3133L). The plugs were then washed once for 30 min at 37 °C with 0.5 mL of 0.5X Tris-Boric acid-EDTA (TBE) buffer and inserted into the wells of a 1% agarose gel (Sigma, MI, USA, Catalog No. A9539). A 50–1000 kb DNA Lambda Ladder (New England Biolabs, MA, USA, Catalog No. N0341S) was loaded as a molecular marker at both ends of the gel. Restriction fragment separation was performed using the CHEF-DR III System (Bio-Rad Laboratories, Hercules, CA, USA) for 22 h at a 120-degree angle and 6 V/cm, with initial and final switch times of 2.16 and 54.17 s, respectively. After staining with ethidium bromide, the bands were visualized under a UV illuminator, and the image was captured using a Gel Doc EZ imager (Bio-Rad Laboratories, CA, USA).

The pulsotypes of *Shigella flexneri* were analyzed using BioNumerics software (version 7.5, Applied Maths, Sint-Martens-Latem, Belgium). A dendrogram was generated using the unweighted pair group method with arithmetic mean (UPGMA), with a 4% Dice coefficient and a 2% tolerance. The criteria for interpreting PFGE patterns were based on the recommendations of Tenover et al. (1995) [35].

#### 2.3.2. Detection of Virulence Genes

DNA was extracted using the DNeasy Blood and Tissue Kit (Qiagen, Hilden, Germany, Catalog No. 69506) following the manufacturer’s protocol for extracting nucleic acid from Gram-negative bacteria. DNA extracts were subjected to PCR amplification for the *ipaH*, *virA*, *ipaBCD*, *ial*, *sen*, *Set1A*, and *Set1B* genes using primers described in a published study [36]. The reaction mixture consisted of 3 µL of DNA, 250 nM of each forward and reverse primer, and 2X GoTaq Master Mix (Promega, Madison, WI, USA, Catalog No. M7123). PCR was performed in a thermal cycler under the following conditions: initial denaturation at 95 °C for 5 min, followed by 30 cycles of 95 °C for 50 s, 55 °C for 1.5 min, and 72 °C for 2 min, with a final extension at 72 °C for 7 min.

## 3. Results

Ten cases were assessed as part of our investigation; however, only four cases were included in our study. The remaining six cases were excluded due to negative culture results for *Shigella* spp. Refer to Table 1 and Table 2 for a summary of the cases and laboratory results, respectively.

### 3.1. Case 1

#### 3.1.1. Patient History

The first case was a 52-year-old male with cerebral palsy, ataxia, epilepsy, right eye strabismus, and aphasia. He was fully dependent on daily care and diapers, and he resided in a rehabilitation center. He presented to the KSUMC ED with a one-day history of high-grade fever (reaching 39.1 °C), loss of appetite, and multiple episodes of foul-smelling, watery, yellowish diarrhea. A medical history was provided by the rehabilitation center staff due to the patient’s non-verbal status. The patient displayed no abdominal pain, hematochezia, or melena. No recent seizure activity was reported.

#### 3.1.2. Examination Findings

At examination, the patient was conscious, alert, and looked well. Vitally, he had a high-grade fever (measuring up to 40.1 °C), tachycardia, and hypotension. Chest auscultation revealed normal heart sounds (S1 and S2) with no murmurs and bilateral equal air entry with no added sounds. The abdomen was soft, lax, and non-tender, with no organomegaly or signs of peritonitis. Rectal digital examination showed no blood.

#### 3.1.3. Laboratory Investigation

Blood Work

An initial complete blood count (CBC) demonstrated leukocytosis with a white blood cell (WBC) count of 19,000/μL, mainly neutrophils. A comprehensive metabolic panel (CMP) revealed a high creatinine level at 127 μmol/L and electrolyte abnormalities, including hypokalemia, hypocalcemia, and hypophosphatemia. The interpretation of venous blood gases (VBG) showed mild metabolic acidosis. Inflammatory markers were elevated with an erythrocyte sedimentation rate (ESR) of 27 mm/h, C-reactive protein (CRP) of 325.320 mg/L, and ferritin level of 2584 mcg/L. Moreover, the laboratory results showed an elevated lactic acid level of 4.32 mmol/L.

Microbiological

Only normal enteric flora was isolated from the XLD agar, CIN agar, Campy agar, and HEK agar plates. However, colorless, small, round, and smooth colonies were noted on the sorbitol MacConkey agar, which were initially presumed to be *E. coli* O157:H7. The culture identified *Shigella* spp. with a 100% confidence level, and it was further speciated as *Shigella flexneri.* The bacteria exhibited resistance to ampicillin, ampicillin–sulbactam, and amoxicillin–clavulanic acid. Additionally, the stool tested negative for *Clostridium difficile* toxin using C. DIFF QUIK CHEK COMPLETE ^®^ (TECHLAB, Inc., Blacksburg, VA, USA) on three different samples collected on different days. Examination of the stool by the parasitology lab revealed no ova or parasites in the concentrated smear.

#### 3.1.4. Imaging Studies

An abdominal X-ray performed by the ED showed dilated bowel loops in the right ilium, raising the possibility of bowel obstruction or toxic megacolon. Due to these concerns, computed tomography (CT) of the abdomen and pelvis was scheduled to rule out bowel ischemia. However, it was delayed due to difficulty in establishing intravenous (IV) access. Eventually, IV-line insertion was successfully achieved by the anesthesia team. The abdominal and pelvic CT scan revealed patent major aortic branches and veins with no signs of bowel ischemia. However, rectal submucosal edema and minimal pelvic free fluid were also shown, findings that were consistent with colitis. To correlate with these imaging findings, a further evaluation via colonoscopy was recommended.

#### 3.1.5. Hospital Course and Management

Upon admission, the patient had a high-grade fever and multiple episodes of severe watery diarrhea that were complicated by acute kidney injury (AKI) and metabolic acidosis. Empiric antibiotics were initiated, including oral vancomycin and IV metronidazole. His renal function parameters and lactic acid level continued to worsen despite receiving IV fluid replacement therapy. Empiric therapy was escalated to IV meropenem due to the patient’s worsening condition, while oral vancomycin was discontinued after ruling out *Clostridium difficile* infection with three negative test results. The patient’s symptoms gradually improved: fever subsided, diarrhea resolved, and lactic acid and renal function parameters normalized.

A diagnosis of *S. flexneri* infection was made through a positive stool culture. The patient completed a one-week course of IV meropenem and was discharged back to the rehabilitation center in stable condition.

### 3.2. Case 2

#### 3.2.1. Patient History

The second case was a 40-year-old male with a background of spastic encephalopathy, spastic diplegia, epilepsy, severe cognitive impairment, poor eyesight, aphasia, and chronic kidney disease of unknown stage, who was brought to KSUMC ED by his caregiver. He was a long-term resident in a rehabilitation center, fully dependent on diapers and his caregiver for daily care. A history was taken from his caretaker, as the patient was non-verbal. He was reported to have a high-grade fever (39.1 °C) for the last three days, which had responded partially to paracetamol. Later, he began to have multiple episodes of foul-smelling, greenish, watery diarrhea, with one episode of rectal bleeding involving fresh blood mixed with stool. He also experienced a loss of appetite, poor oral intake, and lethargy. No respiratory, urinary, or neurological symptoms were reported. It was also reported that other residents were experiencing similar symptoms.

#### 3.2.2. Examination Findings

At a physical examination upon arrival, the patient was conscious and alert, with no signs of distress. He was febrile, with a temperature of 38.8 °C, and tachycardic, with a heart rate of 110–120 beats per minute; otherwise, his other vitals were within normal ranges, including blood pressure, respiratory rate, and oxygen saturation. Normal heart sounds (S1 and S2) with no murmurs and equal bilateral air entry with no added sounds were noted during the chest examination. Abdominal examination revealed a soft, non-tender abdomen with no signs of organomegaly or rebound tenderness. A rectal exam showed fresh blood but no external hemorrhoids or fissures. Limited neurological examination was performed due to the patient’s cognitive impairment; it showed no new focal deficit.

#### 3.2.3. Laboratory Investigation

Blood Work

A laboratory workup performed upon admission revealed leukocytosis of 16,000/μL, low hemoglobin of 128 g/L, and a platelet count within the normal range. CMP showed elevated creatinine (750 µmol/L) and blood urea nitrogen (BUN) (28.7 mmol/L). Metabolic acidosis was demonstrated using VBG with a pH of 7.2 and bicarbonate levels of 12.6 mmol/L. Inflammatory markers levels were elevated with an ESR of 51 mm/h, CRP of 360.330 mg/L, and a procalcitonin level of 4.92 ng/mL.

Microbiological

Short, plump, Gram-negative bacilli were observed in the Gram stain of a positive blood culture bottle, which were subsequently identified as *E. coli* with a 100% confidence level. This isolate was resistant only to ampicillin. In routine bacterial stool cultures, only normal enteric flora was isolated from the XLD agar, CIN agar, Campy agar, and Hektoen agar plates, while colorless, small, round, and smooth colonies were seen on the sorbitol MacConkey agar. They were identified as *Shigella* spp. with a 100% confidence level and speciated as *S. flexneri*. The micro-organism exhibited resistance to ampicillin and aztreonam and intermediate susceptibility to ampicillin–sulbactam and amoxicillin–clavulanic acid. Additionally, the stool tested negative for *Clostridium difficile* toxin using C. DIFF QUIK CHEK COMPLETE^®^ (TECHLAB, Inc., VA, USA) on three separate tests collected on different days. Examination of the stool by the parasitology lab revealed no ova or parasites in the concentrated smear.

#### 3.2.4. Imaging Studies

A chest X-ray performed upon the patient’s presentation revealed minimal left-lower-lobe atelectasis, with mild pericardial effusion and no signs of active pulmonary infection. An abdominal CT scan showed diffuse large bowel wall thickening with adjacent fat stranding, likely related to an infectious/inflammatory process. There was no evidence of obstructive uropathy or renal stones. However, there was an incidental finding of a left undescended testis. During the patient’s hospital stay, a renal ultrasound was conducted and demonstrated small, echogenic kidneys with altered corticomedullary differentiation, consistent with chronic kidney disease. Renal stones and hydronephrosis were not noted.

#### 3.2.5. Hospital Course and Management

The patient was initially managed with aggressive IV fluid resuscitation and sodium bicarbonate to correct the dehydration and metabolic acidosis, alongside close monitoring of electrolytes and renal function. The patient had a fever reaching 39.1 °C; blood cultures were collected, and empirical broad-spectrum antibiotics (IV meropenem and IV vancomycin) were started due to concerns of possible sepsis. During his hospital stay, he developed two additional episodes of hematochezia, for which he was started on proton pump inhibitor infusion. As the blood culture results came back positive for *E. coli*, the antibiotic regimen was modified; meropenem, which he received for 5 days, was de-escalated to ceftriaxone, a third-generation cephalosporin effective against this pathogen, and vancomycin was stopped. The stool culture confirmed *S. flexneri* infection; the patient continued on ceftriaxone for an additional 5 days as a targeted therapy for shigellosis and to complete a 10-day course of IV therapy as a treatment for Gram-negative bacteremia. During his stay, the patient’s diarrhea gradually improved, and he became afebrile with a normal WBC count. His creatinine levels showed a steady downward trend, indicating partial recovery from the acute kidney injury, with the final creatinine measurement recorded at 267 µmol/L prior to discharge. Despite plans for a diagnostic colonoscopy to investigate the cause of rectal bleeding, the procedure was ultimately aborted due to an inability to obtain consent from the patient’s family. The patient’s hematochezia resolved without further invasive intervention, and no additional episodes of bleeding were noted. He was discharged back to the rehabilitation center on an oral course of ciprofloxacin for four additional days, along with iron supplements to treat anemia and sodium bicarbonate for ongoing metabolic acidosis. A nephrology follow-up was scheduled to manage his chronic kidney disease.

### 3.3. Case 3

#### 3.3.1. Patient History

The third case was a 46-year-old male resident of a rehabilitation center with a known case of cerebral palsy, intellectual disability, aphasia, diplegia, type 2 diabetes mellitus, and dyslipidemia. The patient was bedbound, diaper-dependent, and minimally communicative. He presented to the KSUMC ED, where his caregiver reported a one-day history of fever (measuring up to 39 °C) and decreased urine output. He also experienced foul-smelling diarrhea for two days. Similar diarrhea episodes were reported in other individuals at the rehabilitation center; however, there were no specific details mentioned about close contact with those patients.

#### 3.3.2. Examination Findings

At the ED level, the patient was ill-looking, diaphoretic, and at his baseline level of consciousness with minimal communication. He was hypotensive with a blood pressure reading of 82/62 mmHg, tachycardic with a heart rate of 118 beats per minute, and febrile at 39.2 °C. Cardiac examination revealed normal S1 and S2 heart sounds with no murmurs. Upon lung auscultation, there was bilateral equal air entry without any added sounds. Abdominal examination demonstrated a soft, lax, and non-tender abdomen with no signs of rigidity, peritonitis, or organomegaly. No rashes or pressure ulcers were noted on skin examination. Examination of the lower limbs showed bilateral mild pitting edema without signs of deep vein thrombosis (DVT).

#### 3.3.3. Laboratory Investigation

Blood Work

Leukocytosis with a WBC count of 20,000/μL, a normal hemoglobin level of 134 g/L, and a platelet count of 237 × 10⁹/L was demonstrated by the CBC. The patient’s creatinine level (89 μmol/L) was significantly elevated from a baseline of approximately 40 μmol/L, representing a 122.5% increase. Furthermore, his BUN level was high (9.6 mmol/L), and electrolyte results were remarkable for hypokalemia, hypocalcemia, and hypophosphatemia. Arterial blood gas analysis showed a high anion gap metabolic acidosis with a high lactic acid level of 8.8 mmol/L. Additionally, the inflammatory markers ESR, CRP, and procalcitonin were all elevated (23 mm/h, 187 mg/L, and 3.44 ng/mL, respectively).

Microbiological

A midstream urine sample, collected in a sterile, leak-proof container, was processed for urine culture. Cystine-Lactose-Electrolyte-Deficient agar (prepared by Saudi Prepared Media Company, Riyadh, Saudi Arabia) was inoculated using a 0.001 mL loop and incubated at 35 ± 1 °C in an aerobic incubator for 18–24 h. After overnight incubation, 100,000 CFU/mL of large, mucoid, lactose-fermenting colonies were observed on the agar surface. The organism was identified as *Klebsiella oxytoca* with a 100% confidence level. It was found to be an extended-spectrum beta-lactamase producer and multidrug-resistant, exhibiting susceptibility only to carbapenems. Stool culture demonstrated only normal enteric flora from the XLD agar, CIN agar, Campy agar, and HEK agar plates. However, colorless, small, round, and smooth colonies were seen on the sorbitol MacConkey agar. These were identified as *Shigella* spp. with a 100% confidence level. Susceptibility results to antibiotics showed resistance only to ampicillin. Additionally, the stool tested negative for *Clostridium difficile* toxin using C. DIFF QUIK CHEK COMPLETE^®^ (TECHLAB, Inc., VA, USA) on three separate tests collected on different days. Examination of the stool by the parasitology lab revealed no ova or parasites in the concentrated smear.

#### 3.3.4. Imaging Studies

Chest X-rays indicated bilateral pleural fluid effusions without evidence of consolidation or active pneumonia. A repeat point-of-care ultrasound confirmed normal heart function with no pericardial effusion. A lower-limb ultrasound revealed no signs of DVT.

#### 3.3.5. Hospital Course and Management

The patient was initially managed with fluid resuscitation with a one-liter bolus of normal saline; norepinephrine and dopamine infusions were initiated due to persistent hypotension. He was started on broad-spectrum antibiotic coverage with meropenem and vancomycin since he was diagnosed with septic shock, which was suspected to be secondary to the urinary tract infection and concurrent shigellosis.

The patient’s home medications, including metformin and lactulose, were held due to acute kidney injury and the potential for metabolic complications. He received potassium and phosphate supplements to correct the electrolyte imbalances. Prophylactically, the patient was kept on subcutaneous heparin to reduce the risk of venous thromboembolism. Regular vancomycin-level monitoring was performed to ensure therapeutic drug levels and dose adjustments were conducted accordingly.

Throughout his hospital course, the patient gradually improved in terms of his kidney function, blood pressure stabilization without vasopressors, diarrhea episode frequency, and tolerance to oral intake. The laboratory results also showed improvement in WBC count and inflammatory markers. He completed a 7-day course of IV meropenem for the confirmed *S. flexneri* infection. Vancomycin was discontinued due to negative blood cultures and methicillin-resistant *Staphylococcus aureus* surveillance results. Upon discharge, the patient was back to his baseline status and returned to the rehabilitation center in good condition.

### 3.4. Case 4

#### 3.4.1. Patient History

The fourth case was a 38-year-old male with a medical history of congenital cerebral palsy, intellectual disability, dysarthria, and hyperactivity. He was bedbound, used diapers, required daily care, and lived in a rehabilitation center. The patient was brought to the ED by his caregiver with a one-day history of fever (measuring up to 38.3 °C). A history was obtained from the caregiver, as the patient was non-verbal. There was no history of chills, rigors, or respiratory or gastrointestinal symptoms. One week earlier, the patient had presented at the ED with one episode of vomiting; he was diagnosed with aspiration pneumonia and dehydration based on the clinical presentation and chest X-ray findings, which showed right-sided middle-zone consolidation. He was treated with IV amoxicillin–clavulanic acid and discharged on oral formulation to be taken twice daily for seven days. Despite completing the antibiotic course, his fever persisted, prompting his return to the ED. Per the caregiver, there was no known exposure to other sick residents at the rehabilitation facility.

#### 3.4.2. Examination Findings

The patient’s examination showed that he was febrile with a temperature of 38.4 °C and tachycardic. He was normotensive and not tachypneic and maintained an oxygen saturation of 94% on room air. Wheezing and bilateral basal crackles were noted upon chest examination. A cardiac assessment revealed normal S1 and S2 with no murmurs. The abdomen was soft, lax, non-tender, and without guarding, and there was no peripheral edema or signs of DVT.

#### 3.4.3. Laboratory Investigation

Blood Work

The CBC displayed leukocytosis with predominant neutrophils, while the CMP revealed electrolyte imbalances with hypokalemia, with a potassium level of 2.85 mmol/L and mild hypocalcemia (1.8 mmol/L). Lactic acid levels were initially elevated at 5 mmol/L but showed improvement, decreasing to 1.2 mmol/L with appropriate management. Respiratory alkalosis was illustrated via blood gas analysis; the patient was monitored for that matter during his hospital stay. Inflammatory markers were elevated, with an ESR of 47 mm/h, CRP at 442.70 mg/L, and procalcitonin at 31.10 ng/mL. Liver function tests and renal function tests were performed and showed normal results.

Microbiological

Gram-negative rods were visualized under Gram stain from blood culture bottles that were flagged as positive by the instrument. The following day, non-hemolytic smooth white colonies were detected on the sheep blood agar. Medium to large non-lactose fermenting colonies were observed on the MacConkey agar. Subsequently, the organism was identified as *Shigella* spp. with a 100% confidence level and was further speciated as *S. flexneri*. Susceptibility results to antibiotics showed resistance only to ampicillin and intermediate susceptibility to ampicillin–sulbactam and amoxicillin–clavulanic acid. Furthermore, the stool culture was reported as negative for *Shigella* spp.

Other investigations

Electrocardiogram (ECG) findings revealed prolonged QTc intervals, measured at 650 ms and 521 ms during different assessments. Considering the patient’s ECG findings, his antibiotic regimen was changed to avoid QT prolongation-inducing drugs. The ECG abnormality was attributed to both medication side effects and electrolyte imbalances.

#### 3.4.4. Imaging Studies

Chest radiographs revealed bilateral middle-zone consolidations, consistent with the previous diagnosis of aspiration pneumonia.

#### 3.4.5. Hospital Course and Management

Initial management included paracetamol and potassium supplementation. The patient was started on IV cefepime to cover possible pseudomonal infection while considering his QT prolongation and prior amoxicillin–clavulanic acid therapy. Doxycycline and oseltamivir were also initially administered to address potential bacterial and viral causes, respectively. However, oseltamivir was discontinued after respiratory multiplex PCR ruled out viral pathogens. His potassium levels were monitored with regular blood tests and gradually corrected.

Following positive blood culture results for *Shigella flexneri*, cefepime was discontinued after four days of therapy, and he was switched to IV ceftriaxone for three days. The plan was to shift the patient to oral ciprofloxacin following the IV ceftriaxone course. Discharge planning included educating the caregiver to monitor for symptoms and return to the ED if needed. Venous thromboembolism prophylaxis with enoxaparin was maintained throughout hospitalization, considering the patient’s prolonged immobility.

Upon discharge, the patient was stable and afebrile and had improved inflammatory markers, including ESR, CRP, and procalcitonin. He was discharged on oral ciprofloxacin for seven days, with instructions for follow-up if symptoms recurred.

### 3.5. Molecular Diagnosis

The PFGE results showed that all four isolates were indistinguishable and belonged to the same pulsotype (Figure 1). The *ipaH* gene, considered a diagnostic marker for *Shigella* spp., was present in all isolates (Figure 2). These findings align with the microbiological diagnosis of the isolates. Other virulence genes (*virA*, *ipaBCD*, *ial*, *sen*) were only detected in the more invasive strain, causing bacteremia. The *Set1A* and *Set1B* genes were absent in all isolates. The *stx* genes were not tested in this investigation.

### 3.6. Outbreak Management

All confirmed cases of *S. flexneri* were promptly reported to the infection control department at our institution. All isolates underwent further analysis via PFGE. Diapered patients admitted to our hospital with gastroenteritis were placed in contact isolation in accordance with the Centers for Disease Control and Prevention guidelines [37]. Sampling was requested, and strict contact isolation measures were implemented, including assigning single rooms and ensuring proper use of personal protective equipment such as gowns and gloves. Proper donning and doffing of personal protective equipment were emphasized. A contact list was created, recording time, duration, and the personal protective equipment used. Contact isolation signs were placed in patient rooms, and contact isolation cards were used during patient transport. Transfers were restricted or minimized until recovery, except when clinically indicated, in which case, the patient would be the last on the list to ensure thorough terminal cleaning by housekeeping personnel. Disinfectants were used to clean the patient’s environment. Caretakers were not allowed unless necessary, and in such cases, they were educated on hand hygiene and the proper use of protective equipment. Visiting symptomatic patients was restricted. All new *Shigella* spp. cases were tracked to check for connections to the same center, and notifications for suspected cases were made immediately without waiting for results. The Ministry of Health (MOH) was informed of suspected and positive cases. Communication with higher hospital management and the departments overseeing these patients was established. Additionally, education was provided to nursing staff and physicians about the disease and its transmission. Close observation and practice audits, such as adherence to the “5 Moments for Hand Hygiene” and proper use of protective equipment, were emphasized.

## 4. Discussion

This case series addresses the significant challenges implied by *Shigella* spp. infection in institutional settings, particularly among individuals with severe physical and cognitive impairments. There is a scarcity of reported cases of shigellosis outbreaks in long-term healthcare settings. One study reported an outbreak of multidrug-resistant *Shigella sonnei* in a long-stay geriatric nursing center in Queensland, Australia; it highlighted the potential mode of transmission during the outbreak from person to person through shared towels and caring staff, which was facilitated by a lack of infection control precautions in cleaning symptomatic patients [38].

The outbreak in our case series was traced to a small number of cases residing in a rehabilitation center in Riyadh, Saudi Arabia. Four cases were included, and three were confirmed to have *S. flexneri* dysentery through a routine bacterial stool culture. All patients shared the same underlying comorbidities: cerebral palsy, cognitive impairment with minimal communication, and being bedbound. There is a paucity of studies directly linking diapers to *Shigella* spp. outbreaks [39]. However, all of our patients were dependent on diapers, which may have indirectly contributed to horizontal transmission in this case series, likely facilitated by the low infectious dose of *Shigella* spp. (10–100 organisms) [40], and possibly due to poor adherence to standard hygiene practices by the caregivers from the rehabilitation center [39].

The fourth case had an invasive form of *Shigella* spp. infection, that is, bacteremia without any gastrointestinal manifestations. *Shigella* spp. bacteremia is a rare, serious, potentially life-threatening invasive infection likely to occur in elderly, immunocompromised patients or patients with underlying chronic diseases [41,42]. Through molecular studies, the *virA*, *ipaBCD*, *ial*, and *sen* virulence genes were identified in this case, demonstrating the invasiveness of this isolate.

The presence of different virulence genes in *S. flexneri* plays a crucial role in disease severity. *S. flexneri* primarily causes bacillary dysentery through cellular invasion, intracellular replication, and induction of severe inflammation [36,43]. Several studies have demonstrated that key virulence factors, such as *ipaBCD* (bacterial invasion) [44], *virA* (intracellular bacterial spread), and *sen* (enterotoxin production), contribute to disease severity by enhancing epithelial invasion and immune evasion [43]. The *ial* gene, associated with intestinal epithelial cell invasion, has been linked to more invasive infections [36]. The absence of the *Set1A* and *Set1B* genes in all our isolates suggests that enterotoxin production did not significantly contribute to this outbreak. Instead, disease severity was driven by cellular invasion and inflammation, reinforcing the role of invasion-associated virulence factors in pathogenesis [36,43]. Refer to Table 3 for more information about the virulence genes of *Shigella* spp. [36,43].

*S. flexneri* and *S. sonnei* are considered the most common species to cause shigellosis, whereas *S. dysenteriae* is known to cause the most severe form of the disease with high mortality rates [43]. *S. dysenteriae* is well known for its production of Shiga toxin, while *S. flexneri* strains harboring *stx* genes have been reported [45].

Three out of the four laboratory-confirmed cases exhibited signs and symptoms of AKI and metabolic acidosis, which are potentially severe complications of shigellosis, especially in highly susceptible populations, such as those at the extremes of age and immunocompromised patients [46,47]. Shigellosis complicated by AKI is likely associated with severe dehydration due to excessive diarrhea, which can lead to hypovolemia and organ hypoperfusion [48]. AKI can also be a manifestation of HUS, a serious complication that can be triggered by *Shigella* spp. infection, potentially due to the presence of *stx* genes encoding Shiga toxins, which can lead to endothelial damage, microangiopathic hemolytic anemia, thrombocytopenia, and renal injury [36,43,46]. However, none of the three patients in this case series exhibited hemolytic anemia or thrombocytopenia, making HUS less likely. Additionally, *stx* genes were not investigated in this study, so their potential role in these cases remains undetermined. Metabolic acidosis, which is demonstrated by low blood bicarbonate levels and pH alteration, is attributable to gastrointestinal loss resulting from profound diarrhea [49].

For bedbound individuals, especially those residing in long-term healthcare facilities, there is an increased risk of *Shigella* spp. transmission due to close contact with other residents, dependency on caregivers, and difficulties in sustaining personal hygiene [50]. Assistance with basic needs such as bathing and diaper changing is required for bedbound individuals, increasing the risk of *Shigella* spp. transmission if no strict infection control precautions are followed, including proper hand hygiene [51,52].

Infection control and prevention measures and strategies are crucial elements in limiting the spread of *Shigella* spp. infection, particularly in high-risk settings like long-term healthcare facilities [50]. Basic practices, such as proper and frequent handwashing with soap and water or the use of alcohol-based hand hygiene, are effective in preventing infections like *Shigella* spp. from spreading through fecal–oral routes [39,53,54]. In addition, cleaning and disinfecting surfaces, safe food-handling practices, and accessibility to clean water sources are important in preventing transmission [55]. Patient isolation through contact precautions is essential for restraining and controlling the spread of this infection within facilities, although it was clear from the scenario that the infection did not originate from our hospital. Staff training on infection prevention and strict adherence to hygiene protocols, especially at long-term healthcare facilities, profoundly reduce the risk of outbreaks. Wearing personal protective equipment during patient care and limiting contact with the infected individual is also recommended until full recovery. Public health awareness and education through campaigns highlighting the importance of hand hygiene can also augment the prevention efforts [56,57,58].

In our study, PFGE demonstrated robust accuracy in bacterial strain differentiation and outbreak investigation. PFGE analysis revealed that all four isolates shared an identical pulsotype, confirming their genetic relatedness and link to the outbreak. The high discriminatory power of PFGE enables the precise differentiation of closely related strains, outperforming traditional methods like serotyping and biochemical tests [59,60]. This capability is indispensable in distinguishing outbreak-associated *Shigella* strains from sporadic cases, as shown in prior work [61]. Its ability to link clinical isolates to infection sources, such as contaminated food, proved pivotal in tracing outbreaks [62]. Moreover, PFGE provides rapid results within days and facilitates timely public health responses during outbreaks [63]. Similarly, PFGE complements traditional epidemiology by offering molecular evidence to validate transmission hypotheses [64]. These attributes consolidate the utility of PFGE in improving outbreak response and epidemiological understanding. However, it is important to note that PFGE has lower discriminatory power for certain closely related strains than whole-genome sequencing [65]. Additionally, PFGE profile results reveal large DNA fragments and may not detect small genetic changes located on mobile elements, such as gene acquisition or loss, as these changes may not alter the restriction sites or fragment sizes [65,66,67]. Consequently, outbreak isolates may appear indistinguishable by PFGE, while differing in their virulence gene profiles [66,68]. This helps explain the increased virulence observed in the fourth case, which could be due to the acquisition of virulence genes through horizontal gene transfer, a change that PFGE may not detect.

As a limitation, we were unable to investigate the source of this outbreak, as the infection control department in our institution was not part of the task force team for the outbreak investigation at the rehabilitation center. Additionally, we were unable to determine the index case of this outbreak; however, the PFGE analysis revealed the indistinguishable isolates detected from these patients, thus confirming the outbreak. Moreover, we could not assess the full magnitude of the outbreak, as other potential cases presented at different healthcare facilities, whereas our study included only four laboratory-confirmed cases from our institution. This situation highlights the importance of effective communication between institutions and the MOH for the rapid detection and efficient containment of outbreaks.

## 5. Conclusions

This case series presents the challenges of *Shigella flexneri* outbreaks in long-term healthcare facilities, particularly among bedbound individuals with severe cognitive and physical impairments. This outbreak highlights the role of diapers in potential transmission and expands on the severity of potential complications, including AKI and metabolic acidosis. Rapid detection and appropriate antibiotic therapy resulted in favorable outcomes. Effective infection control measures, such as strict hand hygiene, environmental disinfection, and proper staff training, are mandatory in preventing transmission. Although the molecular studies confirmed the outbreak by verifying the genetic relatedness of the isolates, the inability to identify the source or index case necessitates stronger outbreak investigation protocols and oversight in such settings. Strengthening infection prevention strategies is essential to mitigating future outbreaks in vulnerable populations.

## Figures and Tables

**Figure 1 healthcare-13-00971-f001:**
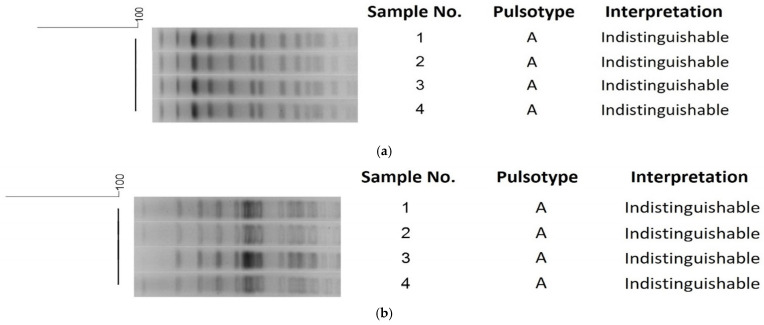
Pulsed-field gel electrophoresis (PFGE) results. (**a**) A dendrogram of NotI-digested *S. flexneri* isolates was constructed based on cluster analysis. DNA relatedness was assessed using BioNumerics v7.5 software (Applied Maths, Sint-Martens-Latem, Belgium). The dendrogram was generated by comparing banding patterns using the unweighted pair group method with arithmetic averages (UPGMA), applying a 4% Dice similarity coefficient and 2% tolerance. Banding pattern interpretation followed Tenover’s criteria [35]. (**b**) A dendrogram of XbaI-digested *S. flexneri* isolates was generated based on cluster analysis. DNA relatedness was analyzed using BioNumerics v7.5 software (Applied Maths, Sint-Martens-Latem, Belgium). The dendrogram was constructed by comparing banding patterns with the unweighted pair group method using arithmetic averages (UPGMA), applying a 4% Dice similarity coefficient and 2% tolerance. Banding pattern interpretation followed Tenover’s criteria [35].

**Figure 2 healthcare-13-00971-f002:**
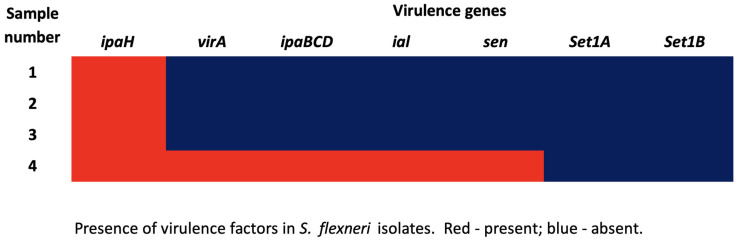
Virulence genes of *Shigella flexneri* isolates.

**Table 1 healthcare-13-00971-t001:** Summary of clinical and demographic characteristics of *Shigella flexneri* cases.

Case No.	Age	Sex	Comorbidities	Type of *Shigella flexneri* Infection	Treatment	Outcome
1	52	Male	Cerebral palsy, ataxia, epilepsy, right eye strabismus, aphasia	Gastroenteritis	IV meropenem for 7 days	Recovered
2	40	Male	Spastic encephalopathy, spastic diplegia, epilepsy, severe cognitive impairment, poor eyesight, aphasia, chronic kidney disease	Gastroenteritis	IV meropenem for 5 days, de-escalated to IV ceftriaxone (total 10-day course), followed by oral ciprofloxacin for 4 days	Recovered
3	46	Male	Cerebral palsy, intellectual disability, aphasia, diplegia, type 2 diabetes mellitus, dyslipidemia	Gastroenteritis	IV meropenem for 7 days	Recovered
4	38	Male	Cerebral palsy, intellectual disability, dysarthria, hyperactivity	Bacteremia	IV ceftriaxone for 3 days followed by oral ciprofloxacin for 7 days	Recovered

**Table 2 healthcare-13-00971-t002:** Summary of laboratory results for *Shigella flexneri* cases.

Lab Findings	Case 1	Case 2	Case 3	Case 4
**CBC**				
WBC	Leukocytosis (19,000//μL)	Leukocytosis (16,000//μL)	Leukocytosis (20,000//μL)	Leukocytosis (20,000//μL)
Hemoglobin	Normal (180 g/L)	Low (128 g/L)	Normal (134 g/L)	Normal (163 g/L)
Platelets	Normal (240 × 10⁹/L)	Normal (158 × 10⁹/L)	Normal (237 × 10⁹/L)	Normal (373 × 10⁹/L)
**Differential**	Predominantly neutrophils	Predominantly neutrophils	Predominantly neutrophils	Predominantly neutrophils
**Inflammatory Markers**				
ESR	Elevated (27 mm/h)	Elevated (51 mm/h)	Elevated (23 mm/h)	Elevated (47 mm/h)
CRP	Elevated (325.320 mg/L)	Elevated (360.330 mg/L)	Elevated (187 mg/L)	Elevated (442.70 mg/L)
Procalcitonin	Not performed	Elevated (4.92 ng/mL)	Elevated (3.44 ng/mL)	Elevated (31.10 ng/mL)
**Renal Function Tests**				
BUN	Normal (6.2 mmol/L)	Elevated (28.7 mmol/L)	Elevated (9.6 mmol/L)	Normal (3.3 mmol/L)
Creatinine	Elevated (127 μmol/L)	Elevated (750 μmol/L)	Normal (89 μmol/L)	Normal (66 μmol/L)
**Electrolytes**				
Potassium	Decreased (3.24 mmol/L)	Normal (4.27 mmol/L)	Decreased (3.44 mmol/L)	Decreased (2.85 mmol/L)
Calcium	Decreased (1.68 mmol/L)	Decreased (1.78 mmol/L)	Decreased (1.95 mmol/L)	Decreased (1.8 mmol/L)
Phosphorus	Decreased (0.34 mmol/L)	Elevated (1.48 mmol/L)	Decreased (0.53 mmol/L)	Decreased (0.68 mmol/L)
Sodium	Normal (138 mmol/L)	Decreased (128 mmol/L)	Normal (136 mmol/L)	Normal (137mmol/L)
**Lactic acid**	Elevated (4.32 mmol/L)	Elevated (2.5 mmol/L)	Elevated (8.8 mmol/L)	Elevated (5 mmol/L)
**ABG/VBG**	Mild metabolic acidosis	Metabolic acidosis	High anion gap metabolic acidosis	Respiratory alkalosis
pH	Decreased (7.234)	Decreased (7.266)	Decreased (7.240)	Elevated (7.405)
pCO_2_	Normal (44 mmHg)	Decreased (30 mmHg)	Elevated (47 mmHg)	Normal (37 mmHg)
HCO_3_⁻	Decreased (17.1 mmol/L)	Decreased (12.6 mmol/L)	Decreased (16.9 mmol/L)	Normal (22.3 mmol/L)
**Liver Function Tests**	Normal	Normal	Normal	Normal

CBC, complete blood count; WBC, white blood cell; ESR, erythrocyte sedimentation rate; CRP, C-reactive protein; BUN, blood urea nitrogen; ABG, arterial blood gas; VBG, venous blood gas; pCO_2_, partial pressure of carbon dioxide; HCO_3_⁻, bicarbonate.

**Table 3 healthcare-13-00971-t003:** Virulence genes of *Shigella* species: functions and pathogenic effects.

Virulence Gene	Function	Effect on Pathogenicity
*ipaH*	Encodes an effector protein secreted via the type III secretion system (T3SS)	Facilitates immune evasion and intracellular survival
*ipaBCD*	Encodes invasion plasmid antigens required for epithelial cell invasion	Essential for bacterial entry into host cells
*virA*	Disrupts host microtubules to promote bacterial spread between cells	Enhances intracellular motility and spread
*ial*	Involved in epithelial cell invasion	Promotes adhesion and penetration into host intestinal cells
*sen*	Encodes enterotoxin (ShET-2) that induces fluid secretion	Causes watery diarrhea by increasing intestinal secretion
*Set1A,* and *Set1B*	Encode Shigella enterotoxin 1	Increases enterotoxin-mediated diarrhea
*stx*	Encodes Shiga toxin (*Stx*), which inhibits protein synthesis in host cells	Causes endothelial cell damage, particularly in the kidneys, leading to hemolytic uremic syndrome (HUS)

## Data Availability

The original contributions presented in this study are included in the article material. Further inquiries can be directed to the corresponding author.

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
