# Peer review of "Shigella flexneri* Outbreak at a Rehabilitation Center: First Report from Saudi Arabia"

_healthcare, 2025, doi:10.3390/healthcare13090971_

Round 1

Reviewer 1 Report

Comments and Suggestions for Authors

Thank you for the opportunity to review this manuscript.

Generally, well-writen. Here are some comments and suggestions to improve the presentation of this case-series.

Line 94: Shigella, change for Shigella spp. across the manuscript

Line 104 : The study design cannot evaluate the effectiveness of treatment. It can only describe the treatment administered and the outcomes, without causality links. Please rephrase without the word effectiveness

Line 112 : Please give a better description of what is suspected Shigella infection...Diarrhea ? Fever ? Did all suspected patients had stool sampled for culture and blood cultures ?

Line 205: you mean negative STOOL results ?

Line 271: isolation for how long ? Was this necessary given the fact that the patient was no longer symptomatic ?

Line 303: Please be consistent with the units used for créatinine and other lab finding across the manuscript. For all cases: When you provide creatinine values could you also provide % change with respect to the baseline values of the case. This could help understand the worsening of creatinin caused by the infection i.e. was there acute to chronic kidney dysfunction. Such details could be added in Table 2

Case 2: no isolation of this case. It would be helpful to say if there is an institutional guideline on infection prevention and control and other isolation measures. Is there a specific procedure for diarrheal diseases ?

Some times in cases description the authors use the present tense and other times the past tense. I would propose to use the past tense uniformally, especially in case description

Line 494: Please avoid commercial names such as Tamiflu 

Lines 507 to 519: In the results please just report what was found and do not give interpretations. This should be done later in the discussion section

Line 525 : it was hospital-acquired because it was acquired in a healthcare institution (the rehabilitation centre), again please refrain from doing such discussions here

Line 527 : change for personal protective equipment. What did this equipment include?

Line 582: diapers is only one aspect of it. Maybe there was a wide spread because these persons were incontinent

Line 600: Add some words on the stx genes and their role in HUS and repeat that these genes were not searched in this study

Line 613: Please moderate this statement. It is well known that alcohol based handrubs are equally effective to water and soap for the vast majority of pathogens. Rare exceptions including Noroviruses and Clostridioides difficile due to alchohol resistance. Otherwise alcohol based solutions should be equally effective.

Are there any epidemiological investigations in the rehabilitation centrre ? Overall number of residents, overall number of residents with symptoms, overall numbers of residents with positive culture. This could allow to formulate an attacke rate. Were there infection prevention and control measures applied to the rehabilitation centre?

Why, although pulse electrophoresis showed identical species, was there a difference in virulence genes in the 4th patient ? Could this be a different isolate ? How do authors explain this difference ?

I would add some sentences on the antimicrobial susceptibility profile of the isolated Shigella spp. Given concerns of AMR rising in these pathogens.

Reviewer 2 Report

Comments and Suggestions for Authors

In this manuscript, the authors reported the first Shigella flexneri Outbreak in Saudi Arabia. Although only four male cases were involved in this study, several key laboratory investigations, such as imaging studies, genetic analysis, and virulence factor analysis, have been included to characterize and demonstrate the pathogen isolates. The experiments were well-designed and performed, and the results were presented effectively. Overall, this manuscript is well-prepared, and the findings could benefit the filed. To move the manuscript forward to publishable level, couple of issues need to be addressed:

  • Shiga toxin is a major virulence factor in Shigella infection, which is often found in S. dysenteriae and even Enterohemorrhagic Escherichia coli but less often found in S. flexneri. Stx gene was mentioned in the text and shown in Figure 2, but it was not fully studied in this manuscript. In this case, I suggest to remove “stx” column in Figure 2 and add some discussion in the text.
  • Only four male cases were included in this study, the number is quite limited. This limitation should be emphasized in the discussion section.
  • English language need to be further polished.
Comments on the Quality of English Language

Can be further improved

Author Response

Please see attachemnt.

Reviewer 3 Report

Comments and Suggestions for Authors

Shigellosis can cause severe morbidity and mortality especially in vulnerable patients with many comorbidities.  In this study, the authors present a case series of patients in healthcare settings and address issues regarding the diagnosis, treatment, and hygiene. The manuscript is well written and present clinical experience regarding shigellosis. Herein are some comments for the authors:

  1. In the discussion part, the authors must comment if there is information in the literature regarding the presence of different virulence genes and a potential different frequency of mortality or efficacy of treatment. The authors have to present a table with the virulence genes and their phenotype action.
  2. The authors have to explain in detail which was “the poor adherence to standard hygiene practices by staff” (line 587). This information can contribute to avoid the same mistakes by other healthcare settings.
  3. Please comment in the discussion part which species of Shigella are more virulent and cause more severe gastroenteritis or invasive infection.
